# Progression of Metastasis through Lymphatic System

**DOI:** 10.3390/cells10030627

**Published:** 2021-03-12

**Authors:** Hengbo Zhou, Pin-ji Lei, Timothy P. Padera

**Affiliations:** Edwin L. Steele Laboratories, Department of Radiation Oncology, Massachusetts General Hospital (MGH) Cancer Center, MGH and Harvard Medical School (HMS), Boston, MA 02114, USA; hzhou20@mgh.harvard.edu (H.Z.); plei3@mgh.harvard.edu (P.-j.L.)

**Keywords:** lymphatic vessel, lymph node, anti-tumor immunity, metastasis

## Abstract

Lymph nodes are the most common sites of metastasis in cancer patients. Nodal disease status provides great prognostic power, but how lymph node metastases should be treated is under debate. Thus, it is important to understand the mechanisms by which lymph node metastases progress and how they can be targeted to provide therapeutic benefits. In this review, we focus on delineating the process of cancer cell migration to and through lymphatic vessels, survival in draining lymph nodes and further spread to other distant organs. In addition, emerging molecular targets and potential strategies to inhibit lymph node metastasis are discussed.

## 1. Introduction

The lymphatic system plays vital roles in both tissue–fluid balance and immunity [1]. In normal physiological conditions, initial lymphatic vessels collect extracellular fluids that carry tissue waste and antigens. These collected fluids and immune cells constitute lymph, which is subsequently transported by collecting lymphatic vessels to draining lymph nodes where adaptive immune responses can be initiated [2]. After being surveilled by lymph nodes, lymph is returned to the blood. Infection, chronic inflammation and cancer can each cause dysfunction in lymphatic vessels [3,4,5] and in draining lymph nodes, which results in impaired immunity [2]. Cancer cells also disseminate through lymphatic vessels and colonize draining lymph nodes. Consequently, metastatic lymph nodes become immune suppressed [6]. Incorporating recent findings in this review, we summarize how cancer manipulates the lymphatic system to form lymph node metastases, avoid the immune system and spread further to other organs, as well as discuss future opportunities for targeting lymph node metastases in the clinic.

## 2. Lymphatic Vessels

Initial lymphatic vessels are the smallest of the lymphatic vessels, and their main job is to absorb interstitial fluid and create lymph. Different from the tight junctions of blood capillary endothelial cells, lymphatic endothelial cells (LECs) in initial lymphatic vessels are oak leaf-shaped and form discontinuous button-like intercellular junctions [7]. The unique microarchitecture of initial lymphatic vessel LECs allows one-way non-partitioned flow of interstitial fluid and its contents into lymphatic vessels, while preventing leak of lymph back into the interstitium [7]. This functional capability arises from overlapping flap structures between adjacent LECs that form primary valves that will open when tissue fluid pressure increases above lymph pressure and close when lymph pressure is higher than the surrounding tissue fluid pressure [8,9,10]. Along with fluid, antigen presenting cells (e.g., dendritic cells (DCs)) can also enter and migrate through initial lymphatic vessels towards the lymph node. It has been demonstrated that this process is dictated by local chemokine CCL21 gradients produced by LECs and interstitial flow [11,12,13,14]. Detailed mechanisms have been discussed in other reviews [15,16,17]. Conceivably, this mechanism can be hijacked by cancer cells to enter the lymphatic system.

Multiple initial lymphatic vessels eventually converge on larger, collecting lymphatic vessels. Unlike initial lymphatic vessels, the LECs in collecting lymphatic vessels are tightly connected through continuous “zipper-like” junctions [7]. Additionally, collecting lymphatic vessels possess one-way intraluminal lymphatic valves and are surrounded by specialized lymphatic muscle cells (LMCs) that cause vessel contraction. The segments between valves behave like individual pumps called lymphangions. Driven by LMC pumping and the one-way valves, fluids and lymphocytes are transported unidirectionally to lymph nodes and eventually to the blood circulation [10,18]. To date, mechanisms by which lymphatic contractility is regulated have been associated with numerous factors, including nitric oxide (NO) and the dynamics of intracellular Ca^2+^ [19,20,21]. After a contraction, NO induced by lymph flow over LECs triggers LMC relaxation and vessel dilation, which allows for lymphangion filling. As flow slows, the short half-life NO signaling fades as NO decays, and the vessel is primed for the next contraction [22]. NO also alters the Ca^2+^ action potentials needed to trigger strong contractions. Intriguingly, our group demonstrated that inducible nitric oxide synthase (iNOS)-expressing CD11b(+)Gr-1(+) cells produce consistent NO, which prevents the temporal NO dynamics needed for stable, autonomous contractions to occur. An infiltrate of iNOS-expressing CD11b(+)Gr-1(+) cells caused by sterile inflammation is able to decrease lymphatic contraction, thereby dampening antigen response [4]. We showed that cancer-associated myeloid-derived suppressor cells (MDSCs) can similarly cripple lymphatic contraction [3].

Of note, in addition to forming vessels for lymph transport, it is becoming increasingly clear that LECs also participate in antigen presentation, which is discussed in other reviews [23,24].

## 3. Lymph Nodes

Lymph nodes (LNs) are critical organs involved with the generation of adaptive immune responses. Each lymph node receives local information from tissues via the lymph, which arrives in LNs via collecting lymphatic vessels [1,2]. LNs have a tissue architecture optimized to allow the interaction of antigen presenting cells with cognate lymphocytes to facilitate the efficient generation of adaptive immune responses. LNs are encompassed by a fibrous capsule with an underlying subcapsular sinus where lymph first enters the lymph node. Some of this lymph will percolate through a conduit system made up of collagen fibers into the cortex of the node, where free antigen can be taken up and presented by resident dendritic cells [1,25]. In addition to free antigen, dendritic cells presenting antigens from the peripheral tissue can also crawl through the floor of the subcapsular sinus to enter the cortex [25]. The remaining lymph drains through the subcapsular sinus around the LN where it arrives in medulla and exits through the efferent lymphatic vessel [26]. The LN parenchyma consists of two primary zones—for the main lymphocyte type that resides there—named the B-cell zone in cortex and T-cell zone in paracortex [25].

Naïve lymphocytes enter the lymph node through specialized blood vessels with characteristic thickened endothelial cells, termed high endothelial venules (HEVs) [25]. Mechanistically, naïve lymphocytes first adhere and roll on HEVs using L-selectin (CD62L) and then associate with numerous glycans on the luminal HEV surface, for instance sialyl Lewis X (sLe^x^) [27]. Subsequently, rolling lymphocytes are activated by chemokines produced by the HEVs, such as CC-chemokine ligand 21 (CCL21) through binding to CC-chemokine receptor 7 (CCR7) on the lymphocyte [28,29]. Of note, CCL21 mRNA is not detectable, whereas CCL21 protein is present within human HEVs [30]. CCL21 mRNA is found in the T-cell zone and lymphatics, which indicates that transcytosis of CCL21 through HEV cells is responsible for lymphocyte recruitment [30]. In addition, shear forces produced by blood flow result in conformational changes of lymphocyte function-associated antigen 1 (LFA1) on lymphocytes, which increases the binding to intracellular adhesion molecules 1 and 2 (ICAM1 and ICAM2) on endothelial cells [31]. Finally, lymphocytes conduct transendothelial migration across the HEV wall into the lymph node where they then follow chemokine paths created by fibroblastic reticular cells (FRCs) [32]. Notably, transendothelial migration does not always occur readily, as it has been observed that some lymphocytes transiently reside in “HEV pockets” [33]. In addition, other immune cells enter the LN via afferent lymphatics, including migratory DCs, memory T cells and effector T cells. Similar to the HEV route, it has been demonstrated that the CCL21-CCR7 signaling axis is critical in regulating immune cell entry through afferent lymphatics, and both CCL19/CCL21-CCR7 and CXCL13-CXCR5 signaling contributes to the intranodal positioning of lymphocytes [25]. CCL1-CCR8 was also shown to be involved in DC homing to the LN parenchyma [34]. 

When a naïve T cell encounters an APC (e.g., DC) carrying cognate antigen in an LN, an interaction between the T-cell receptor (TCR) and peptide-major histocompatibility complex (MHC) occurs (signal 1). CD4 on helper T cells or CD8 on cytotoxic T cells subsequently interacts with the MHC. This interaction can be either transient (<10 min) or long lasting (hours). A long-lasting T cell–APC contact results in robust T-cell activation [35]. Of note, co-stimulatory/-inhibitory molecules (CD80/CD86-CD28/CTLA4, ICAM1/2-LFA-1 and CD2-LFA-3) are also present at the T cell–APC interface, and their interactions (signal 2) are essential for regulating T-cell activation [36]. Activated T cells proliferate, and further differentiate into effector and memory subsets [37].

If after surveilling an LN for several hours, there is no antigen challenge, naïve lymphocytes will exit from the LN through efferent lymphatics to continue systemic surveillance. This egress process is dictated by sphingosine-1-phosphate (S1P) and its receptor, S1PR1 [38]. Lymphocytes expressing S1PR1 sense an intranodal S1P gradient that is low in the parenchyma and high near the medullary lymphatic vessels, and chemotactically migrate towards the higher concentration [38]. Notably, cortical sinuses can play essential roles in the initial steps of lymphocyte entry into medullary sinuses for egress [38]. Disruption of the S1P gradient in LN or decreased S1PR1 expression on lymphocytes can cause compromised egress and immunosuppression [38]. Intriguingly, S1PR1 has also been found on cancer cells [39,40], making it critical to investigate whether cancer cells can utilize the S1P gradient as a means of egress from lymph nodes in order to disseminate further.

Cellular components in LN are diverse, including abundant immune cells, such as B cells, T cells, DCs and macrophages. In addition, stromal cells, such as FRCs, make up a substantial portion of cellular components and help organize the structure of the lymph node. Briefly, B cells are predominantly located in outer portions of the cortex, while T cells and DCs are prevalently found in the paracortical area. Macrophages are separately located in the subcapsular sinus and medulla. Stromal FRCs primarily contribute to the scaffolding of LNs and regulate numerous behaviors, such as the activation and mobilization of local lymphocytes, as well as the prevention of autoimmune reaction [41,42,43]. We discuss how LN cellular components interact with tumor cells during metastasis in the following sections.

## 4. Lymph Node Metastasis

The phenomenon of LN metastases has been long observed [44]. It has been shown that cancer cells can enter lymphatic vessels, migrate to tumor-draining lymph nodes (TDLNs) (Figure 1), grow into lesions in the TDLN and even escape the TDLN to spread to other organs [6,45]. Numerous studies have been conducted to interrogate how mechanistically LN metastasis occurs. Here, we summarize and discuss this complicated process with regard to the most recent discoveries.

### 4.1. Tumor Cell Migration to Draining Lymph Node

Both cell autonomous and non-cell autonomous factors contribute to cancer cell entry into lymphatic vessels. The primary valve structures (button junctions) of initial lymphatic vessel LECs facilitate cell entry of DCs in normal physiology and open when interstitial fluid pressure (IFP) surpasses the internal lymph pressure inside the vessel [7]. Tumors have elevated IFP [46], which can mechanically facilitate the entry of cancer cells into the functional tumor margin lymphatic vessels [5]. Using human melanoma xenografts (A-07 tumor model), Rofstad et al. reported that increased IFP was associated with LN and pulmonary metastasis [47]. Using non-invasive dynamic contrast-enhanced MRI (DCE-MRI), Hompland et al. visually detected that elevated IFP positively correlated with the velocity of fluid outflow from tumors and the incidence of LN metastases in both melanoma and cervical carcinoma xenografts [48]. A recent study by Rofstad et al. demonstrated that human melanoma tumors (R-18 and T-22) that seed LN metastasis exhibit higher IFP and often have a larger hypoxic center, high microvascular and lymphatic vessel density in the tumor periphery, and increased expression of vascular endothelial growth factor A (VEGF-A) and VEGF-C. The authors proposed that the central hypoxia causes the upregulation of proangiogenic factors, which are transported to the tumor edge by outward interstitial fluid flow driven by elevated tumor IFP [49]. As a result, peritumoral lymphangiogenesis is driven by the accumulation of VEGF-C, promoting lymph node metastasis [50]. Even though these studies suggest that high IFP contributes to LN metastasis, direct experimental evidence proves that the causal relation is still lacking. 

The influence of interstitial flow on tumor cell migration is not purely mechanical. A seminal study by Shields et al. revealed that lymphatic drainage-induced interstitial flow can assist human breast cancer cells to establish autologous gradients of secreted CCR7 ligands. The autologous gradients will cause cancer cell chemotaxis in the direction of interstitial flow and toward lymphatic vessels, which promotes LN metastasis [51]. Strikingly, this cellular migration behavior was validated by Polacheck et al. In addition, this work also described a competing CCR7-independent mechanism of autologous chemotaxis where upstream flow causes the phosphorylation of Tyr-397 in focal adhesion kinase (FAK), causing tumor cells to migrate against the flow [52]. Intriguingly, Evje et al. used a computational simulation to interpret how these competing mechanisms—CCR7-dependent autologous chemotaxis and CCR7-independent upstream migration—can both contribute to LN metastasis. The simulations predict that flow-mediated autologous chemotaxis initiates tumor migration towards lymphatics, while upstream migration enables small clusters to detach from the primary tumor [53]. Additionally, Sheih et al. determined that interstitial flow stimulates TGF-β- and MMP-dependent migration of fibroblasts in a 3D invasion assay. As a result, these fibroblasts cause local collagen fiber remodeling in a Rho-dependent fashion and further promote tumor cell invasion [54]. Since cells can actively sense mechanical cues and translate these cues into signaling alterations [55,56,57], it is likely that during tumor evolution, metastatic cancer cells experience multiple mechanical signals mediated by the aberrant lymphatic drainage associated with solid tumors, which ultimately assist dissemination.

In addition to cell autonomous interstitial flow-mediated migration, cancer cells can actively move toward and invade lymphatic vessels utilizing chemokine gradients produced by LECs, which are physiologically designed for leukocyte homing [58]. For instance, CXCL10, CXCL12, CCL1, CCL19 and CCL21 produced by LECs can be sensed by tumor cells expressing cognate receptors and stimulate their chemoattraction towards lymphatic vessels [59]. Furthermore, tumor cells or associated stromal cells are able to secrete factors such as VEGF-C and VEGF-D to induce lymphangiogenesis through VEGFR-3 activation in LECs, thereby providing access to more lymphatic vessels [60]. However, it is under debate whether intratumoral lymphangiogenesis provides more routes for LN metastasis. Intratumor lymphatics vessels do not seem to have lymph flow [5], although they can play roles in regulating immune responses [61]. However, peritumor lymphatics are considered essential for tumor cell delivery to LNs [5,45]. Once in close proximity, tumor cells can associate with and invade lymphatic vessels [62]. Using high-resolution intravital microscopy, it has been visually observed that fluorescently labeled cancer cells intravasate into lymphatic vessels, travel to the subcapsular sinus of LNs and eventually reach the parenchyma of the LN [45,63]. Mechanistically, Kerjaschki et al. propose that tumor cells expressing arachidonate 15-lipoxygenase-1 (ALOX15)—which catalyzes arachidonic acid to 12[S]-hydroxy-eicosatetraenoic acid (12[S]-HETE) and 15(S)-hydroxyeicosatetraenoic acid (15[S]-HETE)—can induce circular defects on lymphatic endothelium, allowing tumor cells to enter the vessel [64]. Moreover, ALOX15 is negatively correlated with metastasis-free survival in patients [64]. 

In preclinic models, mounting evidence demonstrates that epithelial-to-mesenchymal transition (EMT) enhances the motility and invasiveness of cancer cells, promoting lymphatic vessel intravasation [65,66]. Indeed, tumor cells with a high expression of EMT-associated transcription factors, such as Snail 1/2, Twist 1 and ZEB 1, often express high levels of MMPs and form invadopodia to facilitate the degradation of the extracellular matrix. As a result, these cells exhibit a greater capacity to metastasize to LNs [66,67,68,69,70,71]. Interestingly, EMT not only mediates intravasation but also influences chemotaxis during LN metastasis as well. TGF-β-induced EMT can upregulate chemokine receptors, including CXCR4, CXCR5, CCR7 and CCR8 in cancer cells, thus enabling them to follow chemotactic gradients to lymphatic vessels as described above [72]. 

Once inside the draining lymphatic vessels, cancer cells can be transported to LNs by lymph flow [45,63]. Tumor-draining lymphatic vessels display abnormal flow due to the suppression of lymphatic contraction by iNOS^+^/CD11b^+^/Gr-1^+^ myeloid cells [3]. Even though the fluid shear force in lymph flow is dramatically lower than blood flow [73], the impaired contraction may reduce shear stress further and permit higher tumor cell survival [74]. Further, it has been observed that tumor-draining lymphatics can have flow stagnation, allowing cancer cells to associate with each other in the vessel, activate integrin signaling and increase growth and survival in the vessel [75]. Additional investigation on whether LN metastases can be inhibited by modulating lymphatic contraction remains to be performed.

### 4.2. Tumor Cell Survival in Draining Lymph Nodes

#### 4.2.1. Premetastatic Niche

Even though immunosuppression in TDLNs is widely appreciated, mechanistically, it is still not well understood how cancer cells can seed an organ filled with immune cells and escape destruction [76]. In a Lewis lung carcinoma model, EP3^+^/CD11c^+^/SDF-1^+^ DCs in the LN subcapsular regions can be induced by COX-2-derived PGE_2_ even prior to tumor cell infiltration and contribute to the accumulation of Tregs and lymphangiogenesis in tumor-draining LNs. This phenotype can be reversed by COX-2 inhibitors, SDF-1 antagonists and CXCR4 neutralizing antibodies [77]. In stage I/II melanoma patients, RNA levels of IL-10 and IFN-γ, which code for proteins that can induce DCs to express the immunosuppressive enzyme IDO, were discovered to be higher in patients with either cancer-positive sentinel lymph nodes (SLNs) or residual primary tumor, compared to patients with negative SLNs [78]. Similarly, when comparing SLNs to both dormant and inflamed human lymph nodes, it was found that—regardless of metastasis status—SLNs have a reduced cytotoxic T-cell population in melanoma patients [79]. These studies demonstrate that immunosuppression in TDLNs actually occurs prior to metastasis and the pre-metastatic niche emerges before cancer cell arrival [80]. Two major lymphangiogenic factors, VEGF-A and VEGF-C, have been found to induce lymphatic network expansion in sentinel LNs prior to metastasis onset, and contribute to lymphatic metastasis [81,82]. Intriguingly, LECs proliferate and sprout within days of tumor implantation [83]. Transcriptional profiling of LECs in TDLNs reveals that, compared to counterparts in naïve LNs, genes involved in cell division, cell adhesion and immune regulation are upregulated, while transcriptional regulation and differentiation genes are downregulated. This transcriptional signature partially resembles gene expression of LECs in LNs upon viral infection or inflammation caused by ovalbumin (OVA) [83]. In addition, the heparin-binding factor midkine was recently demonstrated to activate mTOR signaling in LECs at distal sites, leading to lymphangiogenesis and pre-metastatic niche formation [84].

Vesicle-associated factors can also condition lymph nodes before tumor dissemination. For instance, exosomes released by melanoma influence extracellular matrix deposition and vascular growth factor expression in TDLNs, and melanoma cells are located in exosome rich regions of TDLNs [85]. Metastatic melanoma cells also have been shown to permanently “educate” bone marrow (BM) progenitors and convert them to a c-Kit^+^/Tier2^+^/Met^+^ pro-vasculogenic phenotype. As a result, these reprogrammed BM progenitors contribute to vascular leakiness and pre-metastatic niche establishment in multiple organs including LNs [86]. Pucci et al. recently reported that melanoma-derived extracellular vesicles (EVs) can spread through lymphatic vessels yet are initially blocked from migrating into the LN by subcapsular sinus (SCS) CD169^+^ macrophages when they arrive in TDLNs. However, with cancer progression, this macrophage barrier is eventually breached and EVs interact with B cells in the LN cortex, resulting in cancer-promoting effects [87]. Lymphatic exudate from melanoma patients was also shown to be rich in extracellular vesicles that can characterize the extent of metastatic spread [88]. Interestingly, most exosome and pre-metastatic niche studies were conducted using melanoma models, raising a question about whether similar events occur universally in other cancers.

#### 4.2.2. Major Histocompatibility Complex (MHC)

One strategy for cancer cells to circumvent immune surveillance in LNs is by suppressing MHC expression so that T-cell recognition is dampened (Figure 2) [89,90]. Indeed, it has been documented that patients bearing cancer cells expressing low MHC I tend to have higher incidence of regional LN metastases, which is associated with low cytotoxic T-cell lysis [91]. Further, in a gastric cancer patient cohort, MHC I was demonstrated to be lower in metastases compared to primary tumors, while PD-L1—a ligand that leads to suppression of T-cell activity—was found to be positively correlated with the number of metastatic LNs [92]. Similar to MHC I, loss of MHC II has been associated with elevated regional LN metastatic burden in cervical, colorectal and other cancers [93,94,95]. 

#### 4.2.3. Tumor-Immune Cell Interaction

After arrival in TLDNs, cancer cells can convert the immune cell landscape to a more permissive environment for cancer growth and survival. LNs contain many different immune cell compartments that create a balance between generating an immune response against pathogens and maintaining tolerance against self-antigen. Different immune cell populations play a spectrum of roles in maintaining this critical balance. These different cell types can be phenotypically changed by cancer cells to promote metastatic growth and survival [6]. For instance, macrophages can be reprogrammed to an M2-like (anti-inflammatory) phenotype, which promote tumor progression [96]. Further, M2 tumor-associated macrophages (TAMs) are involved in lymphangiogenesis and LN metastasis in thyroid, pancreatic and lung cancers [97]. However, depleting all macrophages by blocking G-CSF and CSF1-R also depletes SCS macrophages, which are an important defense against cancer cell entry into LNs [98,99]. The heterogeneity of the macrophage subtypes poses challenges to therapeutic strategies to harness their anti-tumor macrophage properties while suppressing their pro-tumor activities. Intriguingly, based on the high affinity of cationic agarose (C-agarose) with Siglec-1 on the surface of LN SCS macrophages, a recent work used a C-agarose and CpG oligodeoxynucleotides (CpG ODNs) drug delivery system to activate SCS macrophages. Consequently, this treatment shrank both primary tumors as well as LN and lung metastases in a model of human triple-negative breast cancer model [100]. 

DCs—a vital antigen-presenting cell population responsible for T-cell activation—can also be altered by tumor cells to suppress immune activation. Tumor cells can either increase the abundance of a tolerogenic DC subpopulation or cause dysfunction in DCs to induce immunosuppression and facilitate tumor progression [101,102]. In SLNs from breast cancer patients, the inhibition of LN-resident DC activation coincides with an enrichment in immunosuppressive MDSCs and regulatory T cells (Tregs) as well as an increase in T-cell anergy. Moreover, the suppressed state of these LN-resident DCs reduces T-cell effector function [103]. 

Similar to macrophages, a subgroup of tumor-associated neutrophils (TANs) can promote immunosuppression, tumor growth and metastasis [104]. In gastric tumors, the number of CD15^+^ TANs in the primary site and TDLNs is positively correlated with the depth of primary tumor invasion and LN metastasis. These CD15^+^ TANs possibly arise from CXCR2^+^ neutrophils [105]. In a spontaneous mammary carcinoma mouse model, tumor cells were shown to secrete IL-1β to induce IL-17 expression in γδ T cells, which led to the G-CSF-dependent expansion and polarization of neutrophils. This tumor-induced neutrophil expansion and differentiation led to the inhibition of cytotoxic T-cell function and the enhancement of LN metastasis [106]. Moreover, neutrophils aid tumor cell colonization in LNs by releasing leukotrienes, which can be blocked by the pharmacological inhibition of Alox5, a leukotriene-generating enzyme [107]. 

Natural killer (NK) cells also play a role in anti-cancer immune responses. In early head and neck cancer, it has been demonstrated that NK cell cytotoxicity is drastically suppressed when overt LN metastases are present [108,109]. Further, recent melanoma and breast cancer works identified a subset of mature CD56^bright^/CD16^+^ NK cells that infiltrated TDLNs, where their lytic activity was reduced, suggesting that tumor cells suppress NK cell activity in LNs [110,111]. 

Recently, a subset of immunosuppressive B cells was identified, named regulatory B cells (Bregs). Bregs can affect multiple immune cells, including T cells, through the secretion of anti-inflammatory cytokines, such as TGF-β and IL-10, causing inhibited anti-cancer responses [112,113]. More importantly, it was found that Bregs preferentially accumulate in TDLNs across various cancers, including melanoma, gastric, colon and cervical cancers. The depletion of these suppressive Bregs often slowed disease progression and lymph node metastasis [112,114,115,116,117,118]. Compared to B cells, anti-tumor immunity from T cells is intensively studied and well-reviewed [119,120], and it has been acknowledged that cancer can cause T-cell dysfunction [120]. 

Tumor cells are also able to directly influence adaptive immune cells to allow immune evasion. In a melanoma model, tumor cells directly introduced into LNs were rejected by CD8^+^ T cells, whereas when primary tumors were present, T-cell anergy was measured in TDLNs [121]. Moreover, tumor cells implanted in TDLNs after tumor-induced T-cell anergy were able to grow, suggesting that distant CD8^+^ T-cell tolerization can be an initiating event of LN metastasis [121]. Additionally, mounting evidence demonstrates that tumors can cripple T-cell response in TDLNs through diverse mechanisms including, but not limited to, suppressing T-cell sensitization through MDSCs [122], inducing anergic or regulatory CD4^+^ T cells [123,124], directing incomplete differentiation of antigen-specific CD8^+^ T cells [125] and activating Tregs [126]. Collectively, multiple immune compartments can be suppressed or hijacked by cancer cells to allow LN metastasis to grow and survive. It is imperative to identify common factors that are involved in these mechanisms to develop future therapeutic approaches.

#### 4.2.4. Tumor–Stromal Cell Interaction

FRCs are critical LN stromal cells that provide chemokines to help spatially organize LNs, help LNs expand during an immune response [127] and generate the LN conduit network, which facilitates the transport of small antigens to B cells and follicular DCs [128]. FRCs are also altered by tumor cells to facilitate immune evasion. Inflammatory melanoma causes the loss of FRCs and reduced production of CCL21 in secondary lymphoid organs, resulting in the decreased migration of skin-derived DCs and T cells to TDLNs [129]. Similarly, Lewis lung carcinoma also decreases FRC abundance and IL-7 secretion in TDLNs, resulting in reduced T-cell numbers [130]. The authors propose that the tumor-induced loss of FRCs in TDLNs contributes to dampened T-cell responses. In contrast, TDLNs from B16F10 melanoma show FRC proliferation and an expanded stromal network. Analysis of the transcriptional profile of FRCs from non-TDLNs and B16F10 melanoma-draining LNs, however, shows that multiple crucial pathways for leukocyte recruitment, chemokine/cytokine signaling and matrix remodeling are reprogrammed in TDLNs. Consistent with previous findings, CCL21 and IL-7 derived from FRCs in TDLNs are decreased, leading to disorganized immune cell localization and altered cellular composition [131]. Although FRC growth or loss seems tumor specific, most data show that FRCs are reprogrammed by tumor cells to influence normal immune cell composition in TDLNs, and that CCL21 and IL-7 are critical to these changes. These abnormal and disorganized immune cell populations in TDLNs likely impair anti-tumor immunity.

LN LECs are also prone to be stimulated by tumor cells and in turn contribute to cancer dissemination to LNs by lymphangiogenesis and immune regulation, independent of their role in the pre-metastatic niche [81,82]. In vitro co-culture data show that gastric cancer cells induce CXCL1 secretion from LECs, which stimulates lymphatic tube formation in an autocrine manner, possibly through the FAK-ERK1/2-RhoA axis [132]. SCS LECs produce CCL1 and chemoattract cancer cells expressing CCR8 to lymph nodes. Inhibiting CCR8 arrests cancer cells in afferent lymphatic vessels [133]. Moreover, gastric cancer cells also upregulate PD-L1 and IDO expression in TDLN-derived LECs, thereby suppressing inflammatory cytokine production in CD4^+^ T cells [134]. In a B16F10-OVA model, cancer cells release VEGF-C and induce lymphangiogenesis in TDLNs. Further, TDLN-resident LECs cross-present OVA to T cells, which leads to the dysfunctional activation of OVA-specific CD8^+^ T cells [135]. Therefore, tumor-associated LECs can assist metastasis through multiple processes, and targeting tumor–LEC interaction is a therapeutically promising approach.

#### 4.2.5. Metabolic Adaptation

In order to survive, metastatic cancer cells face the challenge of a new metabolic landscape in the metastatic site. Intriguingly, using SCCVII and 4T1 tumor models, our group showed that proliferating tumor cells located in TDLNs do not induce sprouting angiogenesis, but instead utilize the existing lymph node blood vasculature to secure a nutrient supply [136]. However, even after securing a nutrient supply, the metabolic microenvironment is different from the primary site, making metabolic plasticity essential for metastatic cancer cell survival and growth [137]. For example, the reliance on fatty acid metabolism is more important for LN metastasis compared to the primary tumor. Within CD44^bright^ human oral carcinoma cells, a subset of CD36^+^ (fatty acid receptor) cells respond to dietary lipids and increase metastatic initiation in LNs, while blocking CD36 ablates this phenotype [138]. Moreover, Lee et al. demonstrated that B16F10 murine melanoma cells shift their metabolism towards fatty acid oxidation (FAO) when they colonize TDLNs [139]. Mechanistically, the authors proposed that accumulated bioactive bile acids—possibly acting through a vitamin D receptor—induce YAP activation, which upregulates genes mediating fatty acid oxidation [139]. The importance of FA metabolism in LN metastasis was corroborated by Zhang et al., who demonstrated that FABP5 positively correlates with the incidence of LN metastases and poor prognosis in cervical cancer. Further, the data show that the hypoxic TME suppresses miR-144-3p and upregulates FABP5 expression. Consequently, FABP5 induces lipolysis, FA synthesis, NF-κB signaling, EMT and lymphangiogenesis, leading to more LN metastases [140]. Lipid not only provides an energy source but also influences cell membrane fluidity, which is an important factor mediating cell migration [141,142]. Further, the question regarding how much CD36 (lipid uptake) and FABP5 contribute to LN metastasis by enhancing cancer cell entry versus survival in LNs remains to be addressed. Altogether, these data underscore the potential of inhibiting LN metastasis through targeting metabolic pathways involving fatty acid metabolism.

The presence of certain nutrients in lymph also changes the ability of metastatic cancer cells to survive if they subsequently escape to the blood. An investigation by Ubellacker et al. demonstrated that enriched levels of glutathione and oleic acid in lymph modify the levels of unsaturated lipids on the melanoma cell membrane, making the cells less prone to oxidation required for death by ferroptosis [143]. Strikingly, lymph and the LN microenvironment impart ferroptosis resistance to LN metastatic melanoma cells and enhance their ability survive and spread systemically through the blood [143]. 

### 4.3. Lymph Node Metastases Spread Further to Other Organs

The lymph node may not be the final destination of all metastatic cells that arrive there (Figure 2), yet whether and how much LN metastases can contribute to distant metastases is still under debate. Naxerova et al., using somatic variants in hypermutable DNA regions, reconstructed the phylogenetic trees of primary tumors, LN metastases and distant metastases from colorectal cancer patients. Their data showed that in 35% of cases, LN and distant metastases shared a common origin, while in the remaining 65% of cases, distant metastasis appeared to be derived from independent subclones in the primary tumor [144]. Another investigation of the patterns of metastasis in colon cancer patients showed that liver metastasis could originate directly from the primary tumor or from LNMs. These data also showed the LNMs can also seed other LNs, including those in the same regional bed or more proximally (“skip spreading”) [145]. In addition, two back-to-back studies performed genomic analyses using breast cancer patient primary tumors and metastatic lesion samples to build the evolutionary history of the metastatic lesions. These data showed that most of the driver mutations occur in the primary tumor. The authors concluded that axillary LN metastases contribute minimally to distant seeding in breast cancer patients [146,147]. Further, using a Cas9-based method that allows the long-term single-cell lineage tracing of cancer cells in an orthotopic lung cancer model in mice, some distant metastasis was traced back to an origin in mediastinal lymph nodes [148]. Collectively, these data show that in some—but not all—cancer models and patients, LNMs can serve as a source of distant metastasis. These data have been published with a background of multiple clinical trials that have shown that early-stage cancer patients with positive sentinel lymph nodes who are randomized to complete nodal dissection or no further surgery have similar overall survival [149,150,151]. These contrasting sequence-based data and clinical outcomes beg the question as to whether cancer cells in lymph nodes contribute to distant metastases.

To address this pressing question, our group used cancer cells labelled with the photoconvertible protein Dendra2 combined with intravital microscopy to trace the behavior of metastatic cells in LNs. Using 4T1 mammary carcinomas and B16F10 melanomas, this study provided direct evidence that some cancer cells are able to invade lymph node blood vessels and colonize the lung [152]. Investigating lymph node metastases from head and neck cancer patients also showed the presence of cancer cells inside or invading lymph node blood vessels in 7 of 19 patients [152], showing that this mode of distant metastasis can occur in patients. Strikingly, an independent group simultaneously discovered the same phenomenon using distinct approaches. Brown et al. directly introduced cancer cells into afferent lymphatic vessels through microinfusion, and demonstrated that these cancer cells initially accumulated in the SCS of the LN. Subsequently, the cancer cells migrated along LN stromal networks into the center of LN within days, where they invaded blood vessels and disseminated to the lungs [153]. Thus, in some—but not all—cancer models, tumor cells that are metastasized to LNs can invade lymph node blood vessels and colonize distant organs. Nonetheless, the contribution of LN metastases to distant metastases in patients requires further investigation in order to understand the underlying biology of the process and identify which patients are at risk of distant metastasis that originates from lymph node lesions.

## 5. Targeting Lymph Node Metastasis

### 5.1. Controversy over Extensive Resection of Metastatic LNs

Clinically, the therapeutic value of LN status as a prognostic factor for staging is widely appreciated. As discussed, it has only recently become clear that LN metastasis can seed distant metastasis, but that this process only occurs in a fraction of patients. The idea that LN metastases could contribute to distant metastases would suggest the surgical removal of tumor-draining lymph nodes if a sentinel lymph node is positive. However, for any individual patient, there is currently no biomarker to distinguish who is at risk of dissemination from lymph node lesions and who is not. This creates clinically challenging decisions around how metastatic lymph nodes should be treated, as aggressive treatment of lymph nodes is not without significant morbidities and puts patients at risk of developing debilitating lymphedema. The risk of dissemination from lymph nodes needs to be weighed against the risk of lymphedema development for each individual patient. Unfortunately, we currently do not have sufficient evidence to comprehensively inform this decision.

Fortunately, there are some robust clinical data that have changed how lymph node metastases are treated. Large-scale clinical trials in early-stage breast cancer and melanoma patients with disease-positive SLNs report that complete lymph node dissection does not improve overall survival compared to no surgery beyond SLN removal, even though local disease recurrence is reduced [149,150,151]. Similarly, in a retrospective cohort study, lymphadenectomy did not correlate with improved survival in ovarian cancer patients [154]. These data show that for early-stage cancer patients without bulky nodal disease, aggressive nodal dissections should be avoided.

Although, these trials have changed clinical practice, there is still debate about their interpretation. Patients that did not undergo extensive lymph node dissections did receive chemotherapy and radiation, providing an alternative interpretation that non-surgical treatment options are sufficient for these patients [155]. In different trials, the radiation of tumor-draining lymph node beds has been shown to improve outcomes [156,157]. Further, the act of removing the SLN, which theoretically would contain the greatest tumor burden, could itself be therapeutic [150]. It is also possible that the benefits of removing LN metastases are overshadowed by the effects of the surgery itself, which has been shown to lead to cancer progression in some settings [158,159,160]. In addition, TDLNs may retain impaired yet residual functional anti-tumor immunity, which could provide some tumor-suppressive activity [161,162]. Therefore, LN dissection may remove metastases along with immune functions that could be exerting a systemic effect on distant metastatic disease. Further exploration of the ability of radiation and chemotherapy to prevent disease progression from lymph nodes while preserving immune function and limiting the risk of lymphedema is warranted.

### 5.2. Strategies to Treat LN Metastases

#### 5.2.1. Targeted LN Surgical Removal

Surgery is still routinely adopted to treat LN metastases in the clinic. To conserve non-cancerous LNs and avoid trauma, a clip can be placed on suspicious LNs during diagnostic ultrasonography-guided biopsy and cytologic evaluation. Positive LNs can then have radioactive iodine ^125^I seeds placed on them to direct surgical removal (γ-detection probe), while non-cancerous LNs can be spared [163]. This clip placement method has been shown to decrease the SLN surgery false negative rate after neoadjuvant chemotherapy in breast cancer patients [163,164,165]. Further, to avoid the use of radioactive isotopes, a clinic report demonstrated that magnetic seeds can serve as a promising substitute to mark LN metastases, which can subsequently be localized by magnetometer-guided intraoperative identification [166]. Further, to minimize the extent of surgery, clinical factors, such as age, tumor location, tumor stage, and ER/PR, status can be integrated with mammography, in order to construct a predictive nomogram for axillary LN metastases in breast cancer patients [167].

#### 5.2.2. Targeted LN Agent Delivery

One method to eliminate cancer cells in LNs would be to locally enrich therapeutic agents. However, because of the anatomic and physiological features of the lymphatic system, it is difficult to achieve a high drug concentration in LNs through conventional intravenous delivery approaches [168,169]. To overcome this obstacle, nanosized carriers can be used to increase the efficiency of drug delivery to LNs [170]. Even though the cargo of small-scale carriers can vary greatly—including siRNA, cytotoxic agents or immunogens—the fundamental delivery approaches can be categorized as passive or active [170,171]. 

The passive approach utilizes the size or chemical characteristics of the particle to enhance transport to LNs. For example, pegylated liposomal doxorubicin—doxil—is used clinically and has shown efficacy in regional LNs in gastric carcinoma patients when delivered through submucosal injection [172]. Further, solid lipid nanoparticles have been developed to enhance drug stability in the body [173]. Interestingly, Cabral et al. recently demonstrated that sub-50 nm polyethylene glycol (PEG)-based micelles containing platinum anti-cancer agents (DACHPt/m) can accumulate and inhibit melanoma LN metastases even when delivered through blood [174]. By comparing the targeting efficiency of DACHPt/m with doxil (80 nm) and larger micelles (70 nm), the authors found that bigger particles do not exhibit the same anti-metastatic efficacy. Thus, particle size (below 50 nm) dictates the ability to extravasate from blood vasculature and penetrate LN metastases [174]. This concept was corroborated by Liu et al., who managed to reduce the nanoparticle size to 5 nm and found that this size reduction enhances drug deposition in TDLNs [175]. In addition to size, the surface charge of the drug carrier also plays crucial roles. OVA-conjugated cationic hydrogel nanoparticles can elicit stronger B cell expansion and CD4^+^ T-cell activation in lung DLN, compared to anionic particles, which can be attributed to the preferred association of resident DCs with cationic particles [176]. 

In addition to liposomes and hydrogel nanoparticles, there are other promising nanoparticles for targeting LNs. Dendritic polymers (dendrimers) drain into peripheral lymphatic capillaries after subcutaneous injection, showing their potential to be used for lymphatic imaging and LN drug delivery [177]. Dendrimer drainage and retention in LNs, as well as drug release, can be modulated by changing the dendrimer size, hydrophilicity and surface charge [177,178,179]. Polypropylene sulfide (PPS) nanoparticles, when injected interstitially, can be internalized by DCs in dLNs [180]. Through this targeted method, antigens and adjuvants can be delivered to DCs resident in dLNs [181,182,183]. Further, small LN-targeting nanoparticles (25 nm) can activate complement and cause DC maturation, which results in antigen-specific adaptive immunity [184,185]. Seminal work by Schudel et al. demonstrated that, by using thiol-reactive oxanorbornadiene (OND) linkers, the cargo release can be controlled in nanoparticles with PPS cores. OND linkers have half-lives that can be programmed from hours to days, and by this programmable payload release, the access of drug to different lymphocyte subpopulations can be enhanced in dLNs [186]. 

In contrast to passive approaches, active approaches harness the binding of particles to a specific target in the LN. By adding LyP-1—a 9-amino acid cyclic peptide—to the surface of doxorubicin-loaded liposomes, greater binding was measured to MDA-MB-435 metastatic lesions in popliteal and iliac LNs, thereby reducing disease burden [187]. Similarly, liposomes conjugated with a F(ab’)2 fragment (Thy1.1 ligand) or IL-2 have been found to prevent adoptively transferred T cells in LNs from immunosuppression when intravenously injected, which extends the efficacy of adoptive cell therapy [188]. APCs, like DCs, can also be targeted by nanoparticles. Cruz et al. discovered that subcutaneous vaccination with pegylated polylactic-co-glycolic acid (PLGA) nanoparticles conjugated with TLR-3 and -7 ligands or monoclonal antibodies targeting CD40, DEC-205 and CD11c can improve internalization by DCs, generating greater CD8^+^ T-cell activation [189]. Subcutaneous implantation of a C-agarose hydrogel containing CpG oligodeoxynucleotides (ODNs) was demonstrated to decrease LN and lung metastatic burden in both mammary carcinoma and melanoma models [100]. C-agarose possesses strong affinity to Siglec-1 (CD169) expressed on SCS macrophages, and the cargo—CpG ODNs—is synthetic DNA fragments with unmethylated CpG motifs that can activate TLR9. As a result, the C-agarose+CpG complex can stimulate SCS macrophages and elicit anti-tumor immunity against LN metastases [100]. Another active delivery method uses genetically engineered *Salmonella typhimurium* A1-R, a strain auxotrophic for arginine and leucine. These bacteria can specifically target tumor cells and ablate axillary and popliteal LN metastases when administered through lymphatic vessels [190].

### 5.3. Potential Targets in LN Metastases

To date, there are no therapeutics specifically designed to treat LN metastases. However, multiple signaling pathways are attractive targets, including (i) VEGFC/D-VEGFR-3, (ii) CCL19/CCL21-CCR7, (iii) CXCL12-CXCR4, (iv) COX2-PGE2 and (v) lipid metabolic enzymes. Below, we outline the current progress with a focus on targeting lipid metabolism.

#### 5.3.1. Targeting Lymphangiogenesis

Even though there is evidence demonstrating that VEGFA can induce lymphangiogenesis and drive LN metastasis, its primary influence is on angiogenesis [191]. The VEGFC/D-VEGFR-3 signaling axis plays an essential role in lymphangiogenesis, and it has been validated by numerous studies that blocking this signaling pathway can prevent LN metastasis in animal models [192]. However, multi-kinase inhibitors, such as sunitinib and sorafenib, impacting both VEGFR-2 and VEGFR-3, show limited efficacy in patients with metastatic disease when delivered orally [193,194]. In animal models of spontaneous lymph node metastasis, multi-kinase inhibitors of both VEGFR-2 and VEGFR-3 did not impair lymphatic metastasis when orally given after lymph node seeding and primary tumor removal [195]. Consistent with these data, LN metastasis was shown to not depend on angiogenesis for growth, highlighting a mode of failure for anti-angiogenic therapy [136]. Notably, these small molecule inhibitors are promiscuous, targeting PDGRs, c-KIT, FLT3 and RET, in addition to VEGFRs, which confounds the interpretation of these results [193]. Moreover, soluble factors, such as PDGF-BB, HGF, bFGF, S1P, angiopoietins (ANGs) and TGF-β, can contribute to lymphangiogenesis [192], bypassing VEGFR-3 inhibition. Recent work by Gengenbacher et al. developed a murine metastatic melanoma model that possesses a sustained functional intratumoral lymphatic network, which contributes to seeding peripheral metastases [196]. In this model, the authors demonstrated that LECs are addicted to Ang2-Tie2 signaling for lymphatic vessel maintenance. Presurgical Ang2 neutralization regressed intratumoral lymphatics and improved survival [196]. Thus, to inhibit lymphangiogenesis, targeting pathways in addition to VEGFC/D-VEGFR-3 signaling are likely required.

#### 5.3.2. Targeting Chemotaxis

The CCL19/21-CCR7 axis is important in DC and lymphocyte homing to LNs. However, this pathway can be hijacked by cancer cells to facilitate their migration to LNs [197]. In the pre-clinical setting, siRNA silencing and antibody blockade of CCR7 have demonstrated promising results in inhibiting metastasis [198]; however, an inhibitor for clinical use is still lacking. Similarly, the CXCL12-CXCR4 axis can contribute to LN metastasis [197] and maybe a relevant target for LN metastasis, though this has not been specifically tested. However, the CXCR4 antagonist, plerixafor (AMD3100), has been clinically approved for hematopoietic stem cell mobilization from bone marrow, and other agents, including monoclonal antibodies, aptamers and RNA interference, are also under development [199]. Owing to their great anti-tumor and anti-metastatic potential, CXCR4 antagonists have been tested clinically in multiple cancers, such as acute myeloid leukemia (AML), glioblastoma (GBM), breast, pancreatic and prostate cancers [198], and the preliminary results are promising. In a phase I study, the intravenous administration of balixafortide (a CXCR4 antagonist) in combination with eribulin chemotherapy was well-tolerated and showed encouraging responses in patients with pretreated and relapsed HER2-negative metastatic breast cancer [200]. In a phase IIa study, the combination of motixafortide (a CXCR4 antagonist) and pembrolizumab, when administered intravenously, enhanced the outcomes of chemotherapy in metastatic pancreatic ductal adenocarcinoma [201]. One caveat of targeting chemokine receptors is the great redundancy existing among different receptors, which can lead to compensatory resistance in cancers. Further, blocking chemotaxis would target the additional spread of the cancer, but may not affect disease that has already metastasized to lymph nodes and other sites. This constraint limits the patient populations in which a strategy of blocking chemotaxis would yield positive results.

#### 5.3.3. Targeting Lipid Metabolism

Mounting evidence suggest that lipid metabolism is an attractive target to inhibit LN metastases, including targeting of arachidonic acid and its metabolites. First, COX2 catalyzes the conversion of arachidonic acid to PGE2, a powerful inflammatory regulator that contributes to LN metastasis through immunosuppression [77]. Second, in tumor cells, ALOX15 catalyzes the conversion of arachidonic acid to 12[S]-HETE as well as 15[S]-HETE, and these metabolites create “holes” in lymphatic endothelium facilitating tumor cell entry [64]. In addition, Alox5 in neutrophils induces leukotriene production (from arachidonic acid) and contributes to metastatic initiation [107]. Finally, two recent studies revealed that to better adapt to the LN microenvironment, tumor cells reprogram their metabolism and increase their reliance on fatty acid metabolism, including enhanced lipolysis, fatty acid uptake, fatty acid synthesis and fatty acid oxidation [138,139,140]. It is important to explore how arachidonic acid production and consumption are influenced by this metabolic switch.

Multiple compounds can affect lipid metabolism, but in the context of cancer, one of the most clinically tested class of drugs is the statins. Statins decrease cholesterol levels by suppressing 3-hydroxy-3methylglutaryl-CoA reductase (HMGCR), which is the mevalonate pathway rate-limiting enzyme. To date, statins have been tested as a monotherapy or in combination with other agents in multiple cancers, including lung, esophageal, liver, pancreatic, colorectal, prostate, breast and endometrial cancer [202]. Exciting progress has been reported in unresectable hepatocellular carcinoma, pancreatic ductal adenocarcinoma, AML, refractory or relapsed ovarian cancer, prostate cancer and non-small cell lung cancer, which are thoroughly reviewed by Hassanabad [202]. Celecoxib, a COX2 inhibitor initially approved for osteoarthritis and rheumatoid arthritis as an analgesic and anti-inflammatory agent, has been widely examined in clinic trials due to its strong anti-tumor potential. Promisingly, positive outcomes in patients have been demonstrated in head and neck, breast, prostate and colorectal cancer [203]. To date, numerous agents have shown encouraging results in preclinical models and are under development to target fatty acid uptake, synthesis, mobilization, storage and modification, in order to fine-tune lipid metabolism [204]. The effect of statins, celecoxib and other lipid metabolism modulators on LN metastases should be comprehensively evaluated as these compounds advance in their clinical development.

## 6. Concluding Remarks

The TDLN is a critical site to elicit anti-tumor immunity. However, because of direct drainage, TDLNs can also be colonized by metastatic cancer cells and cause immunosuppression. Mounting evidence demonstrates that LN metastases can also serve as a source of hematogenous dissemination. Thus, the treatment of LN metastasis can potentially decrease systemic metastatic burden. However, it is critical to identify biomarkers that can distinguish which LN metastases are prone to spread distantly in order to prevent overtreatment. The removal of lymph nodes can impair systemic anti-tumor immune responses and leave the patient prone to lymphedema. Thus, the decisions surrounding how to treat an individual patient with metastatic lymph nodes are challenging and critical. It is imperative that research into lymph node metastasis continues in order to provide improved information on which to base these clinical decisions. Further, additional tools to treat lymph node metastasis, including targeting lymphangiogenesis, chemotaxis and lipid metabolism, could generate great therapeutic benefits. The ultimate goal of the treatment of metastatic lymph nodes is to eliminate the ability of the disease to spread from lymph nodes while preserving systemic anti-cancer immune function and preventing lymphedema. These are difficult, but critical, challenges for the field.

## Figures and Tables

**Figure 1 cells-10-00627-f001:**
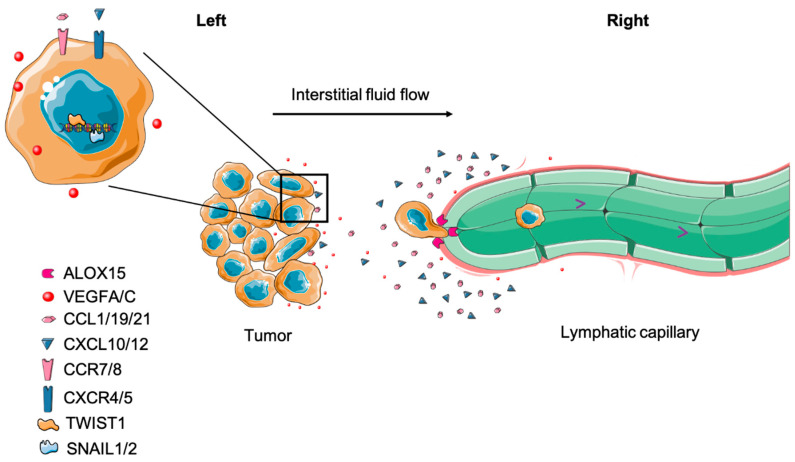
Cancer cells migrate and invade lymphatic vessels. (**Left**) Primary tumor cancer cells release VEGF A/C to promote local lymphangiogenesis. Cancer cells also undergo EMT to increase their invasiveness. In addition, they can upregulate CCR7/8 and CXCR4/5 expression, enabling their responsiveness to the corresponding ligands, CCL1/19/21 and CXCL10/12, respectively. (**Right**) In response to these stimuli, cancer cells migrate to lymphatic vessels through chemotaxis. Some cancer cells express ALOX15, which catalyzes arachidonic acid to 12[S]-HETE and 15[S]-HETE. Accumulation of these metabolites causes circular defects on lymphatic endothelium, facilitating cancer cell entrance. Of note, interstitial fluid flow not only mechanically contributes to cancer cell migration to lymphatic vessels but also assists cancer cells to produce and follow autocrine chemokine gradients to lymphatic vessels.

**Figure 2 cells-10-00627-f002:**
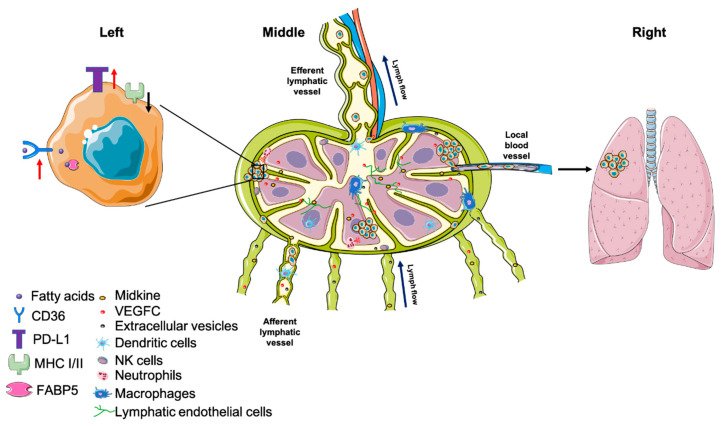
Cancer cells survive in lymph nodes and spread to other organs. (**Left**) To adapt to the lipid-rich environment in lymph nodes, cancer cells enhance their reliance on fatty acid metabolism by upregulating CD36 and FABP5. In addition, to avoid immune destruction, cancer cells increase expression of the immunosuppressive checkpoint molecule PD-L1, while decreasing MHC I/II expression. (**Middle**) Midkine and VEGF-C secreted by cancer cells induce lymphangiogenesis in draining lymph nodes prior to seeding, generating a pre-metastatic niche. Similarly, cancer cells also release extracellular vesicles to interact with B cells in the cortex, promoting lymph node metastasis. Although the lymph node is filled with various immune cell populations, the microenvironment is immunosuppressive. Tolerogenic DCs, M2-like TAMs, and CD15+ TANs were enriched in metastatic lymph nodes. These cells contribute to inhibition of cytotoxic T cells. The lytic activity of NK cells is also reduced in metastatic lymph nodes. (**Right**) After surviving in the lymph node, cancer cells can invade the lymph node blood vasculature and further spread to distant organs. Cancer cells can also exit through the efferent lymphatic vessel.

## Data Availability

No new data were created or analyzed in this study. Data sharing is not applicable to this article.

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
