# Peer review of "Progression of Metastasis through Lymphatic System"

_cells, 2021, doi:10.3390/cells10030627_

Round 1

Reviewer 1 Report

The authors present a comprehensive overview of the current understanding of mechanisms and consequences of lymphatic metastasis, together with valuable insights and reflections on current and future therapeutic interventions. Some adjustments and changes to the text are however needed. Listed below.  

Scientific content

Overall a very interesting overview, covering diverse aspects of lymphatic metastasis and cancer-induced immunological changes in the tumour-draining lymph nodes.

Minor edits:

Line 82: The LN cortex consists of two primary zones—named for the main lymphocyte type that resides there—the T-cell zone and B-cell zone [26]. This statement is incorrect. Suggest exchanging cortex to parenchyma. (B-cell zone=cortex; T-cell zone=paracortex, ref. 26.)

Line 88-90. The authors here refer to mouse studies only and should make that clear for the reader. Human HEVs do not express CCL21, but it is only provided by surrounding FRCs. This is a major difference between the two species, which may have implications for changes in disease. Ref: 10.1182/blood-2004-11-4353. Carlsen et al. 

Line 116-117: Lymphocytes expressing S1PR1 sense an intranodal S1P gradient that is low in the parenchyma and high near the medullary lymphatic vessels, and chemotactically migrate towards the higher concentration [38]. The Cyster lab, in the referenced paper, shows that lymphocytes exit through cortical sinuses, which are distinct from medullary sinuses. Should be corrected.

Repetitive information on ferroptosis (line 228/230 and again line 414/415) citing the same single reference both times without adding more insight in the latter mentioning.

Line 109- Of note, costimulatory molecules (CD80/CD86-CD28/CTLA4, 109 ICAM1/2-LFA-1 and CD2-LFA-3) are also present at the T cell-APC interface and their 110 interactions (signal 2) are essential for regulating T cell activation. CTLA-4 is not co-stimulatory. Either re-phase the sentence to immunomodulatory or co-stimulatory/-inhibitory or only list the CD28.

Line 302-303. Add information on in which tumour type(s) the response was seen (ref. 101). Same thing Line 394-395, ref 142.  As the authors also point out, a lot of research has been done on melanoma, which may have specific mechanisms to interact with the lymph node. It is of interest to the reader to know about the tumour type for the studies.

Illustrations

The illustrations fill their purpose.

In Fig 2, besides LN blood vessels, the alternative route for distant metastatic spread through the lymphatic vasculature should be outlined.

Missing perspectives

As in most reviews on this topic, the authors discuss the VEGF/VEGFC contribution to LN metastasis. Most data are based on ectopic overexpression of the ligands in fast-growing transplantable tumour models, with limited, if any, clinical relevance, which is also reflected in the failure of targeting VEGFRs in clinical trials, as the authors also discuss. Line 196-197 and section Targeting lymphangiogenesis. Line 591-604.

Here the authors could include data, implicating other pathways, in tumour-induced lymphangiogenesis and metastasis. Lymphangiogenesis is not only driven by the VEGFR pathway but can also be promoted by other mechanisms including e.g. ANG2/TIE2 E.g. DOI: 10.1158/2159-8290.CD-20-0122.  Earlier work have also implicated PDGFbb, FGF2 and HGF (reviewed in 10.1172/JCI71606). It would be of high interest to include a reflection around this in the section Targeting lymphangiogenesis.

References

In general, there is an adequate and well-balanced use of references, however, I have some comments.

1.Introduction: The authors self-site three of their previous reviews that all covers the same topic (lymphatic metastasis), Ref. 1, line 17, ref 3 and 7, line 26, for a section that makes very general statements and that for statements would also be covered by other reviews cited in the text. It can be questioned why all these 3 reviews need to be included. (Ref 3 is only cited at one more place in the text, again for a general statement). Would be recommended to revise this.

2. Line 43, ref 12-16. Reference 13 is a review and could better be included, together with ref. 17 and 18, line 44, where the authors specifically reference reviews that cover the topic.

3. Line 291-292. Reference 96 is not correct for this statement. It describes heterogeneity in the immune system and which techniques can be used to study it but not cancer-induced phenotypic changes of immune cells. Should be exchanged.

4. Line 352. For a one-sentence statement, the authors use the three different reviews by the same last author (Turley SJ), covering the same topic: reference 128,129 and 130. None of these at any other place in the text. Should be revised.

Author Response

We appreciate reviewer’s input. By incorporating these comments, the quality of this manuscript will be greatly improved. Please see our responses to the comments point-by-point (comments in black and our responses in red). Revision in manuscript is highlighted in yellow.

Reviewer 1

Comments and Suggestions for Authors

The authors present a comprehensive overview of the current understanding of mechanisms and consequences of lymphatic metastasis, together with valuable insights and reflections on current and future therapeutic interventions. Some adjustments and changes to the text are however needed. Listed below.  

We thank the reviewer for their helpful comments.

Scientific content

Overall a very interesting overview, covering diverse aspects of lymphatic metastasis and cancer-induced immunological changes in the tumour-draining lymph nodes.

Minor edits:

Line 82: The LN cortex consists of two primary zones—named for the main lymphocyte type that resides there—the T-cell zone and B-cell zone [26]. This statement is incorrect. Suggest exchanging cortex to parenchyma. (B-cell zone=cortex; T-cell zone=paracortex, ref. 26.)

Thank you for your input. This sentence has been corrected. Please see line 82-83.

Line 88-90. The authors here refer to mouse studies only and should make that clear for the reader. Human HEVs do not express CCL21, but it is only provided by surrounding FRCs. This is a major difference between the two species, which may have implications for changes in disease. Ref: 10.1182/blood-2004-11-4353. Carlsen et al. 

An explanation of the murine and human difference has been added as well as suggested reference. Please see line 90-93.

Line 116-117: Lymphocytes expressing S1PR1 sense an intranodal S1P gradient that is low in the parenchyma and high near the medullary lymphatic vessels, and chemotactically migrate towards the higher concentration [38]. The Cyster lab, in the referenced paper, shows that lymphocytes exit through cortical sinuses, which are distinct from medullary sinuses. Should be corrected.

According to referenced paper, there are two modes of egress supported by evidence. Lymphocytes can egress through cortical sinuses then medullary sinuses. Alternatively, some cells can pass directly from the subcapsular sinus to the through medullary sinuses. We acknowledge the importance of cortical sinuses and have adjusted our description. Please see line 124-125.

Repetitive information on ferroptosis (line 228/230 and again line 414/415) citing the same single reference both times without adding more insight in the latter mentioning.

To avoid, repetitiveness, this ferroptosis study has been removed from former section-“Tumor cell migration to TDLN”. Please see deletion of text in line 235.

Line 109- Of note, costimulatory molecules (CD80/CD86-CD28/CTLA4, 109 ICAM1/2-LFA-1 and CD2-LFA-3) are also present at the T cell-APC interface and their 110 interactions (signal 2) are essential for regulating T cell activation. CTLA-4 is not co-stimulatory. Either re-phase the sentence to immunomodulatory or co-stimulatory/-inhibitory or only list the CD28.

We have adopted co-stimulatory/-inhibitory expression as suggested. Please see line 115.

Line 302-303. Add information on in which tumour type(s) the response was seen (ref. 101). Same thing Line 394-395, ref 142.  As the authors also point out, a lot of research has been done on melanoma, which may have specific mechanisms to interact with the lymph node. It is of interest to the reader to know about the tumour type for the studies.

We have specified the tumor models used in above description. Ref. 101 (now ref 100, in line 314) used the human triple-negative breast cancer cell line, MDA MB 231. Ref. 142 (now ref 139, in line 409) used the murine melanoma line, B16F10.

Illustrations

The illustrations fill their purpose.

In Fig 2, besides LN blood vessels, the alternative route for distant metastatic spread through the lymphatic vasculature should be outlined.

Cancer cell dissemination from efferent lymphatic vessel has been added to illustration. Please see Figure 2 image and legend in page 10.

Missing perspectives

As in most reviews on this topic, the authors discuss the VEGF/VEGFC contribution to LN metastasis. Most data are based on ectopic overexpression of the ligands in fast-growing transplantable tumour models, with limited, if any, clinical relevance, which is also reflected in the failure of targeting VEGFRs in clinical trials, as the authors also discuss. Line 196-197 and section Targeting lymphangiogenesis. Line 591-604.

Here the authors could include data, implicating other pathways, in tumour-induced lymphangiogenesis and metastasis. Lymphangiogenesis is not only driven by the VEGFR pathway but can also be promoted by other mechanisms including e.g. ANG2/TIE2 E.g. DOI: 10.1158/2159-8290.CD-20-0122.  Earlier work have also implicated PDGFbb, FGF2 and HGF (reviewed in 10.1172/JCI71606). It would be of high interest to include a reflection around this in the section Targeting lymphangiogenesis.

Thank you for your comments and suggestions. We have added new discussion of the clinic failure of VEGFR inhibitors in Targeting lymphangiogenesis section. We have now included alternative pathways that allow tumors to bypass VEGFR inhibition and induce lymphangiogenesis. Please see line 622-630.

References

In general, there is an adequate and well-balanced use of references, however, I have some comments.

1.Introduction: The authors self-site three of their previous reviews that all covers the same topic (lymphatic metastasis), Ref. 1, line 17, ref 3 and 7, line 26, for a section that makes very general statements and that for statements would also be covered by other reviews cited in the text. It can be questioned why all these 3 reviews need to be included. (Ref 3 is only cited at one more place in the text, again for a general statement). Would be recommended to revise this.

Ref 1 is a more comprehensive review discussing lymphatics roles in diseases including but not limited to cancer. Even though ref 3 and 7 are overall about lymphatic metastasis and they share perspectives on tumor-immune cell interactions and immune suppression, ref 7 also discusses polyclonality of lymph node metastases. In the revised introduction we have removed the citation for Ref 7. Only Ref 3 (now ref 6) is cited for statement about the immune suppression of metastatic lymph nodes. Please see line 26.

2. Line 43, ref 12-16. Reference 13 is a review and could better be included, together with ref. 17 and 18, line 44, where the authors specifically reference reviews that cover the topic.

Citation has been adjusted following your suggestion. Please see line 43-44.

3. Line 291-292. Reference 96 is not correct for this statement. It describes heterogeneity in the immune system and which techniques can be used to study it but not cancer-induced phenotypic changes of immune cells. Should be exchanged.

This citation has been exchanged with review from our group discussing tumor growth and immune evasion. Please see line 302.

4. Line 352. For a one-sentence statement, the authors use the three different reviews by the same last author (Turley SJ), covering the same topic: reference 128,129 and 130. None of these at any other place in the text. Should be revised.

Latest summary of FRCs roles in LN is cited, while other two reviews are removed. Please see line 365.

Reviewer 2 Report

This review is very well written by international experts in the field. The topic, which describes the role of the lymphatic network in metastatic progression, is a relatively unexplored subject. This journal article is therefore original and important.

Just a few recommendations: In order to increase the readability of the article, some additional figures would be welcome. For example, a figure summarizing the anatomy of the lymphatic network would be beneficial.

Regarding the paragraphs on targeting the lymphatic network, a few words on the route of drug administration (subcutaneous, intramusculaire or intradermal vs orally or intravenous) would be appropriate.

Author Response

We appreciate reviewer’s input. By incorporating these comments, the quality of this manuscript will be greatly improved. Please see our responses to the comments point-by-point (comments in black and our responses in red). Revision in manuscript is highlighted in yellow.

Reviewer 2

Comments and Suggestions for Authors

This review is very well written by international experts in the field. The topic, which describes the role of the lymphatic network in metastatic progression, is a relatively unexplored subject. This journal article is therefore original and important.

We thank the reviewer for the kind and supportive comments.

Just a few recommendations: In order to increase the readability of the article, some additional figures would be welcome. For example, a figure summarizing the anatomy of the lymphatic network would be beneficial.

We appreciate reviewer’s suggestion. Anatomy of lymphatic network has been well presented by reviews from our and other groups. Given this work is focused on the steps required for lymphatic metastasis, we prioritize the illustration of cell populations, signaling factors and processes involved in this process.

Regarding the paragraphs on targeting the lymphatic network, a few words on the route of drug administration (subcutaneous, intramusculaire or intradermal vs orally or intravenous) would be appropriate.

Delivery routes had been described in “Targeted LN agent delivery” section. As you suggested, administration routes of VEGFRs and CXCR4 inhibitors have been specified in “Targeting lymphangiogenesis” and “Targeting chemotaxis” sections. Please see line 587-590, 615, 617, 643, and 647.